PLoS ONE RESEARCH ARTICLE

# Differential asthma odds following respiratory infection in children from three minority populations

Eric M. Wohlford[1], Luisa N. Borrell[2]*, Jennifer R. Elhawary[1], Brian Plotkin[1], Sam S. Oh[1], Thomas J. Nuckton[1], Celeste Eng[1], Sandra Salazar[1], Michael A. LeNoir[3], Kelley Meade[4], Harold J. Farber[5], Denise Serebrisky[6], Emerita Brigino-Buenaventura[7], William Rodriguez-Cintron[8], Rajesh Kumar[9], Shannon Thyne[10], Max A. Seibold[11], José R. Rodríguez-Santana[12], Esteban G. Burchard[1,13]

1 Department of Medicine, University of California, San Francisco, CA, United States of America,
2 Department of Epidemiology & Biostatistics, Graduate School of Public Health & Health Policy, City University of New York, New York, NY, United States of America, 3 Bay Area Pediatrics, Oakland, CA, United States of America, 4 Children's Hospital and Research Center Oakland, Oakland, CA, United States of America, 5 Department of Pediatrics, Section of Pulmonology, Baylor College of Medicine and Texas Children's Hospital, Houston, TX, United States of America, 6 Pediatric Pulmonary Division, Jacobi Medical Center, Bronx, NY, United States of America, 7 Department of Allergy and Immunology, Kaiser Permanente-Vallejo Medical Center, Vallejo, CA, United States of America, 8 Veterans Caribbean Health Care System, San Juan, Puerto Rico, United States of America, 9 The Ann and Robert H. Lurie Children's Hospital of Chicago, Chicago, IL, United States of America, 10 Department of Pediatrics, University of California, Los Angeles, Los Angeles, CA, United States of America, 11 National Jewish Health, Denver, CO, United States of America, 12 Centro de Neumologia Pediátrica, Caguas, Puerto Rico, United States of America, 13 Department of Bioengineering and Therapeutic Sciences, University of California, San Francisco, CA, United States of America

* Luisa.Borrell@sph.cuny.edu

**Data Availability Statement:** Biological, environmental and phenotypic data analyzed in the current study are available in the dbGAP repository (study accession numbers: SAGE phs00921.v1.p1,

## Abstract

### Rationale

Severe early-life respiratory illnesses, particularly those caused by respiratory syncytial virus (RSV) and human rhinovirus (HRV), are strongly associated with the development of asthma in children. Puerto Rican children in particular have a strikingly high asthma burden. However, prior studies of the potential associations between early-life respiratory illnesses and asthma in Puerto Rican and other minority populations have been limited.

### Objectives

We sought to determine whether early-life respiratory illness was associated with asthma in Puerto Rican, Mexican American, and African American children.

### Methods

Using a logistic regression analysis, we examined the association between early-life respiratory illnesses (report of upper respiratory infection (URI), pneumonia, bronchitis, and bronchiolitis/RSV) within the first two years of life and physician-diagnosed asthma after the age of two in a large cohort of Puerto Rican, Mexican American, and African American children.

GALA II phs001180.v1.p1). Psychosocial data analyzed in the current study (experience of discrimination and socioeconomic status) are not publicly available due to the sensitive nature of the data and privacy concerns for study participants. Psychosocial data is currently stored in the UCSF Box repository and is available upon reasonable request at asthma.collaboratory@ucsf.edu

**Funding:** This work was supported in part by the Sandler Family Foundation, the American Asthma Foundation, the RWJF Amos Medical Faculty Development Program, Harry Wm. and Diana V. Hind Distinguished Professor in Pharmaceutical Sciences II, National Institutes of Health R01HL117004, R01HL128439, R01HL135156, 1X01HL134589, R01HL141992, 5T32GM007546, 1U01HL138626-01A1, National Institute of Health and Environmental Health Sciences R01ES015794, R21ES24844, the National Institute on Minority Health and Health Disparities P60MD006902 RL5GM118984, R01MD010443 and the Tobacco-Related Disease Research Program under Award Number 24RT-0025, 27IR-0030.

**Competing interests:** The authors have declared that no competing interests exist.

**Abbreviations:** RSV, respiratory syncytial virus; HRV, human rhinovirus; GALA II, Genes-Environments and Admixture in Latino Americans; SAGE, The Study of African Americans, Asthma, Genes and Environments; URI, upper respiratory infection; SES, socioeconomic status.

## Measurements and main results

While early-life respiratory illnesses were associated with greater asthma odds in Puerto Ricans, Mexican Americans, and African Americans, these associations were stronger among Puerto Rican children. Specifically, in Puerto Ricans, the odds was 6.15 (95% CI: 4.21–9.05) if the child reported at least one of the following respiratory illness: URI, pneumonia, bronchitis or bronchiolitis. The odds were also higher in Puerto Ricans when considering these conditions separately.

## Conclusions

We observed population-specific associations between early-life respiratory illnesses and asthma, which were especially significant and stronger in Puerto Ricans. Taken together with the known high burden of RSV in Puerto Rico, our results may help explain the high burden of asthma in Puerto Ricans.

## Introduction

Asthma is the most common chronic disease in children [1, 2], with genetic, environmental, and infectious risk factors. [3–5] Though the global burden of asthma is increasing, certain racial/ethnic and geographic populations are at especially high risk. Puerto Ricans are among the most severely affected populations in the world. [4] Approximately 36.5% of Puerto Ricans report they currently or previously had asthma, compared to 9.4% of African Americans, 7.6% of non-Hispanic whites, and 7.5% of Mexican Americans. [6, 7] These striking differences extend to asthma morbidity and mortality, which are 2.4- and 4-fold higher in Puerto Ricans compared to whites, respectively. [6, 7] Asthma is a complex disease and often presents as a function of genetic, environmental, and socio-cultural risk factors. [8–13] Yet asthma, and more specifically asthma disparities, are still not well understood. We propose to examine a single, but significant, facet of asthma that may explain differences in asthma prevalence across minority populations: early-life respiratory illnesses.

Several epidemiological studies have established a strong association between the development of childhood asthma or recurrent wheeze with exposure to severe, early-life respiratory illnesses across populations. [14–24] Associations with asthma were strongest for infections caused by respiratory syncytial virus (RSV) [14, 17, 20–24] and human rhinovirus (HRV), leading to a 2.6 and 9.8 fold probability of asthma, respectively. [18–20] Before one year of age, persistent wheezing illness is most commonly caused by RSV, which later changes to HRV in older children. [25] Both RSV and HRV are known to cause bronchiolitis, which can be a severe respiratory infection in children and is linked to later asthma development. [22, 26] These viruses have a complex interaction between genetics and environmental exposures in determining risks for asthma and related outcomes. [27, 28] Additionally, Puerto Rico has an RSV season that is year-round whereas the mainland United States only reports a 20-week season (**Fig 1**). [29, 30]

It is unclear at this time if differential responses to early-life respiratory illnesses contribute to the striking asthma disparities seen across minority populations. It is possible that genetic predisposition, environmental influences, and early-life respiratory illness work together to increase asthma susceptibility in high-risk populations. Our aim was to investigate the association of early-life respiratory illnesses with asthma susceptibility seen in our large and well-

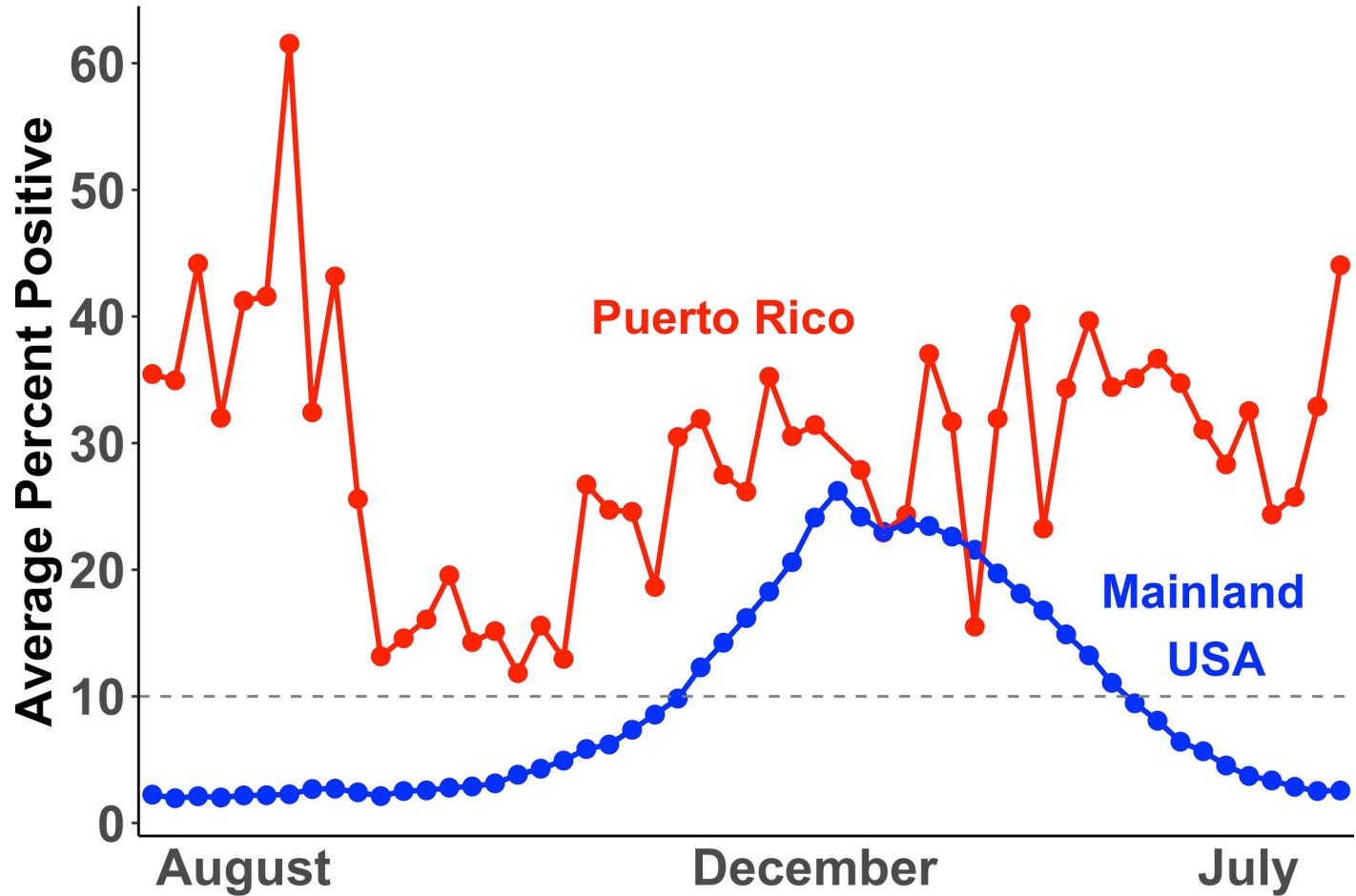

**Fig 1. Respiratory syncytial virus (RSV) season in Puerto Rico (red) and the mainland United States (blue).** RSV season begins when 10% or more of RSV tests are positive. Data shown are a simplified representation of the data obtained from McGuinness et al. Pediatr Infect Dis J. 2014. [29]

phenotyped cohort of minority children. In light of disparate disease prevalence, [6, 29, 30] we examined this association in each population (Puerto Ricans, African Americans and Mexican Americans), separately and combined. We hypothesize that differences in the prevalence of clinically diagnosed early-life respiratory infections such as upper respiratory infections (URI), pneumonia, bronchitis, and bronchiolitis across populations may be associated with asthma prevalence and further these associations may vary across these three minority population.

## Methods

### Study population

The study included participants recruited through two studies: the Genes-Environments and Admixture in Latino Americans (GALA II) study and the Study of African Americans, Asthma, Genes and Environments (SAGE), described in detail elsewhere. [10, 13, 31] Briefly, GALA II and SAGE are parallel case-control studies of asthma conducted between 2006 to 2014 in Latino (Mexican American and Puerto Rican) and African American children aged 8–21, respectively. SAGE participants were recruited from the San Francisco Bay Area and GALA II participants were recruited from across the continental United States (Chicago, Houston, New York City, and the San Francisco Bay Area) and Puerto Rico. Questionnaires

were administered to adult participants and parents of minors. All participants provided written consent to being in the study. Consent was obtained from all adult participants and parent/legal guardians of minor participants. The study protocols for both GALA II and SAGE were approved by the UCSF Human Research Protection Program Institutional Review Board (IRB) and all institutions participating in recruitment obtained the appropriate approvals from their IRBs for recruitment related activity.

Out of a total of 6,023, participants with asthma onset prior to the age of two (n = 1,450) and those individuals without complete illness and covariate data (n = 2,301) were excluded from the study. Those diagnosed with asthma before age two were initially excluded from analysis to better delineate any causal association between early-life respiratory illness before age two and subsequent diagnosis of asthma. The exclusion criteria yielded an analytical sample of 2,824 subjects, including both cases (n = 1,091) and controls (n = 1,733).

### Outcomes

The outcome of our study was physician diagnosed asthma diagnosed after the age of two reported by the parent. [31] Eligible control subjects had no reported history of asthma, lung disease, or chronic illness, and no reported symptoms of coughing, wheezing, or shortness of breath in the two years before enrollment.

### Exposure

The exposure was parental report of physician diagnosed early-life respiratory illnesses within the first two years of life. Early-life respiratory illnesses included URI, pneumonia, bronchitis, and bronchiolitis/RSV, which were analyzed separately. In addition, the report of at least one of the four previous illnesses was used to categorize children with "any respiratory illness" and none otherwise. As this was a retrospective study, we could not rely on pathogen identification as a means to confirm the viral species present during the illness.

### Covariates

Consistent with previous studies, [32, 33] we considered factors known to be associated with early-life respiratory illnesses and with asthma. These included sex, underweight at birth (yes/no), maternal smoking during pregnancy (yes/no), whether the subject was breastfed (yes/no), number of older siblings (none, one, two or more), socioeconomic status (SES; low, medium, high; described further in **S1 Text**), and recruitment site. However, recruitment site was not included in the African American models because all the participants were recruited from one site only.

### Statistical analysis

Descriptive statistics were calculated for the overall study population and for each minority population separately. We used logistic regression to quantify the association between physician-diagnosed asthma and self-reported respiratory illnesses in the first two years of life in all three populations (Puerto Ricans, Mexican Americans, and African Americans) combined. In addition to sex, underweight at birth, maternal smoking during pregnancy, breastfeeding, number of older siblings, SES and recruitment site, we adjusted for the underlying substructure of populations in the dataset using global African and European ancestry estimates (described further in **S2 Text**) as a proxy. Because race/ethnicity may act as an effect modifier due to the large differences in asthma prevalence, [6, 7] we estimated this association stratified

by race/ethnicity. Significance for main effects was determined at $p < 0.05$. The statistical programming language R version 3.5.1 was used to perform all analyses.

## Data availability

Biological, environmental and phenotypic data analyzed in the current study are available in the dbGAP repository (study accession numbers: SAGE phs00921.v1.p1, GALA II phs001180.v1.p1). Psychosocial data analyzed in the current study (experience of discrimination and socioeconomic status) are not publicly available due to the sensitive nature of the data and privacy concerns for study participants. Psychosocial data is currently stored in the UCSF Box repository and is available upon reasonable request at asthma.collaboratory@ucsf.edu.

## Results

Descriptive characteristics of our final study population are presented in **Table 1**. When compared with children without asthma, those with asthma were more likely to be male, to have a mother who smoked during pregnancy, and have lower SES. These distributions were observed in our sample regardless of race/ethnicity with a few exceptions. For instance, while

**Table 1. Descriptive statistics for selected characteristics for GALA II and SAGE participants: 2006–2014.**

| | Puerto Rican | | | | Mexican American | | | | African American | | | | Total Population | | | |
|---|---|---|---|---|---|---|---|---|---|---|---|---|---|---|---|---|
| | *Case* | | *Control* | | *Case* | | *Control* | | *Case* | | *Control* | | *Case* | | *Control* | |
| | N | % | N | % | N | % | N | % | N | % | N | % | N | % | N | % |
| Number of subjects | 284 | 100 | 783 | 100 | 375 | 100 | 504 | 100 | 432 | 100 | 446 | 100 | 1091 | 100 | 1733 | 100 |
| Males | 150 | 52.8 | 361 | 46.1 | 212 | 56.5 | 210 | 41.7 | 218 | 50.5 | 187 | 41.9 | 580 | 53.2 | 758 | 43.7 |
| Underweight at birth | 32 | 11.3 | 61 | 7.8 | 24 | 6.4 | 35 | 6.9 | 36 | 8.3 | 45 | 10.1 | 92 | 8.4 | 141 | 8.1 |
| In-utero smoke exposure | 14 | 4.9 | 40 | 5.1 | 14 | 3.7 | 9 | 1.8 | 78 | 18.1 | 54 | 12.1 | 106 | 9.7 | 103 | 5.9 |
| Breastfed | 141 | 49.6 | 432 | 55.2 | 285 | 76 | 401 | 79.6 | 262 | 60.6 | 244 | 54.7 | 688 | 63.1 | 1077 | 62.1 |
| Number of older siblings | | | | | | | | | | | | | | | | |
| *0* | 74 | 26.1 | 200 | 25.5 | 132 | 35.2 | 195 | 38.7 | 247 | 57.2 | 208 | 46.6 | 453 | 41.5 | 603 | 34.8 |
| *1* | 92 | 32.4 | 274 | 35 | 138 | 36.8 | 176 | 34.9 | 101 | 23.4 | 132 | 29.6 | 331 | 30.3 | 582 | 33.6 |
| *2 or more* | 118 | 41.5 | 309 | 39.5 | 105 | 28 | 133 | 26.4 | 84 | 19.4 | 106 | 23.8 | 307 | 28.1 | 548 | 31.6 |
| Socioeconomic status* | | | | | | | | | | | | | | | | |
| *High* | 89 | 31.3 | 271 | 34.6 | 114 | 30.4 | 116 | 23 | 163 | 37.7 | 160 | 35.9 | 366 | 33.5 | 547 | 31.6 |
| *Medium* | 51 | 18 | 136 | 17.4 | 79 | 21.1 | 84 | 16.7 | 113 | 26.2 | 137 | 30.7 | 243 | 22.3 | 357 | 20.6 |
| *Low* | 144 | 50.7 | 376 | 48 | 182 | 48.5 | 304 | 60.3 | 156 | 36.1 | 149 | 33.4 | 482 | 44.2 | 829 | 47.8 |
| Recruitment site | | | | | | | | | | | | | | | | |
| *Chicago* | 18 | 6.3 | 23 | 2.9 | 137 | 36.5 | 180 | 35.7 | - | - | - | - | 155 | 14.2 | 203 | 11.7 |
| *Houston* | 1 | 0.4 | - | - | 96 | 25.6 | 91 | 18.1 | - | - | - | - | 97 | 8.9 | 91 | 5.3 |
| *New York* | 27 | 9.5 | 34 | 4.3 | 20 | 5.3 | 64 | 12.7 | - | - | - | - | 47 | 4.3 | 98 | 5.7 |
| *San Francisco Bay Area* | 1 | 0.4 | 1 | 0.1 | 122 | 32.5 | 169 | 33.5 | 432 | 100 | 446 | 100 | 555 | 50.9 | 616 | 35.5 |
| *Puerto Rico* | 237 | 83.5 | 725 | 92.6 | - | - | - | - | - | - | - | - | 237 | 21.7 | 725 | 41.8 |
| URI | 57 | 20.1 | 35 | 4.5 | 24 | 6.4 | 27 | 5.4 | 67 | 15.5 | 16 | 3.6 | 148 | 13.6 | 78 | 4.5 |
| Pneumonia | 13 | 4.6 | 5 | 0.6 | 20 | 5.3 | 11 | 2.2 | 24 | 5.6 | 14 | 3.1 | 57 | 5.2 | 30 | 1.7 |
| Bronchitis | 36 | 12.7 | 10 | 1.3 | 37 | 9.9 | 10 | 2 | 16 | 3.7 | 6 | 1.3 | 89 | 8.2 | 26 | 1.5 |
| Bronchiolitis/RSV | 31 | 10.9 | 14 | 1.8 | 6 | 1.6 | 4 | 0.8 | 6 | 1.4 | 2 | 0.4 | 43 | 3.9 | 20 | 1.2 |
| Any Listed | 89 | 31.3 | 54 | 6.9 | 75 | 20 | 46 | 9.1 | 98 | 22.7 | 31 | 7 | 262 | 24 | 131 | 7.6 |

*Socioeconomic status was derived from a combination of mother's education level, health insurance status, and household income weighted by region, see **S1 Text** for more information.

Puerto Rican and Mexican American children without asthma were more likely to be breastfed than their counterparts with asthma, the opposite was true for African Americans. Additionally, we found that Puerto Ricans were 4- to 9-fold more likely to report an RSV infection or bronchiolitis in the first two years of life than Mexican Americans and African Americans, respectively (**Fig 2**).

In our total population, we observed that the odds of asthma was 4.31 (95% CI: 3.15–5.96) for URI, 2.66 (95% CI: 1.67–4.30) for pneumonia, 7.04 (95% CI: 4.44–11.60) for bronchitis, 5.82 (95% CI: 3.26–10.80) for bronchiolitis/RSV, and 4.50 (95% CI: 3.52–5.78) if the participant reported having at least one early-life respiratory illness (**Table 2** and **Fig 3**). In the population-specific stratified analyses, we also observed that the odds of asthma in Puerto Ricans was 5.25 (95% CI: 3.34–8.37) for URI, 7.23 (95% CI: 2.66–23.0) for pneumonia, 13.0 (95% CI: 6.51–28.20) for bronchitis, 7.27 (95% CI: 3.83–14.50) for bronchiolitis/RSV, and 6.15 (95% CI: 4.21–9.05) if the participant reported having at least one early-life respiratory illness (**Table 2** and **Fig 3**). Similarly, and with few exceptions, we observed significant associations between early-life respiratory illness and asthma in Mexican American and in African American children. However, these associations were smaller in magnitude than the ones observed among Puerto Ricans. Specifically, the odds of asthma in Mexican Americans and African Americans,

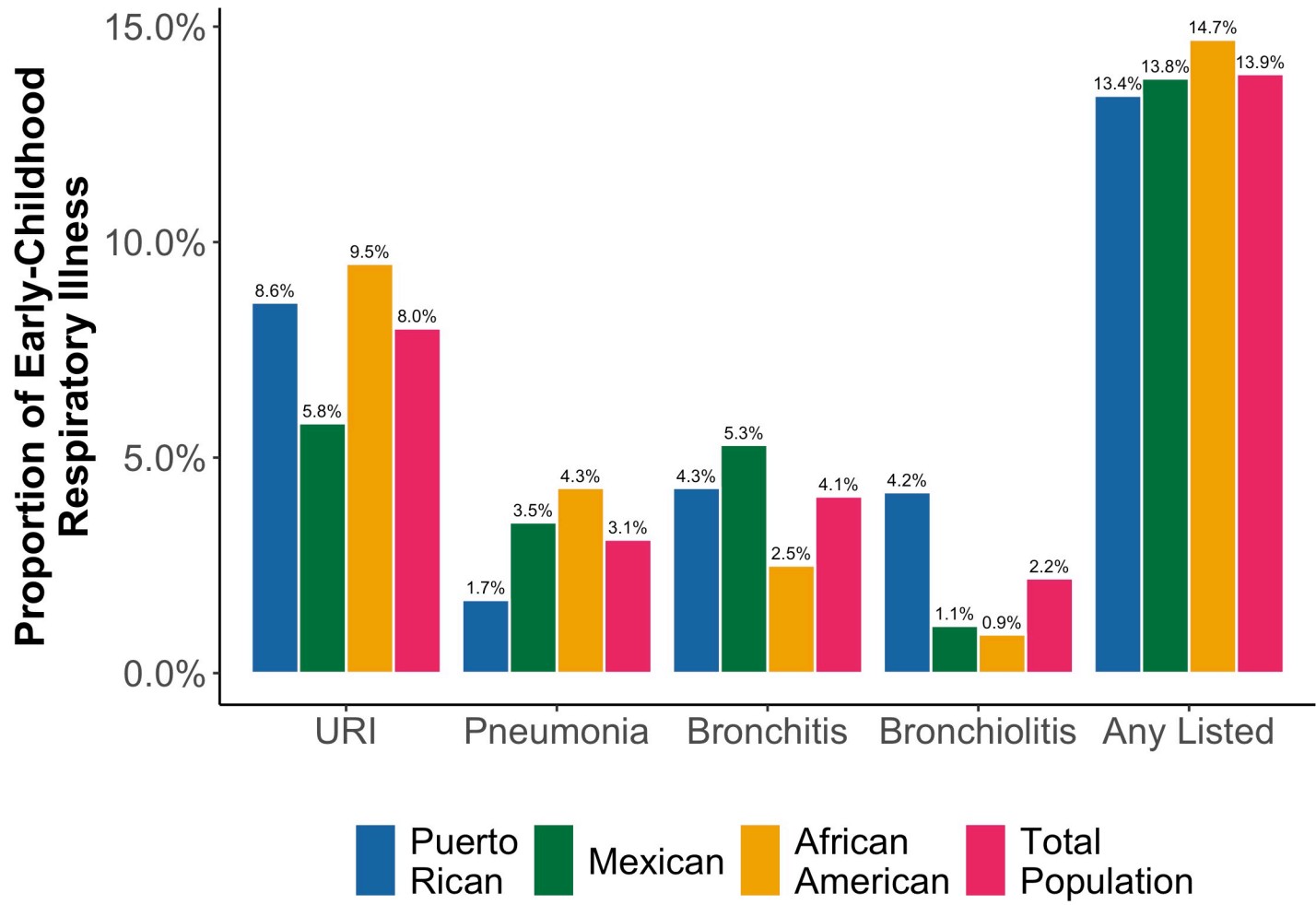

**Fig 2. Population-specific proportions of selected respiratory illnesses during the first 2 years of life for our study population.**

**Table 2. Adjusted\* odds ratios and confidence intervals for the association analysis between respiratory illnesses in the first two years of life and physician-diagnosed asthma after the age of two according to race/ethnicity and in the total population in GALA II and SAGE: 2006–2014.**

|  | Puerto Rican (n = 1,067) | Mexican American (n = 879) | African American (n = 878) | Total Population (n = 2,824) |
|---|---|---|---|---|
| URI | 5.25 (3.34–8.37) | 2.17 (1.12–4.24) | 4.77 (2.76–8.71) | 4.31 (3.15–5.96) |
| Pneumonia | 7.23 (2.66–23.0) | 2.05 (0.96–4.61) | 1.90 (0.97–3.86) | 2.66 (1.67–4.30) |
| Bronchitis | 13.0 (6.51–28.20) | 4.78 (2.39–10.40) | 2.70 (1.08–7.71) | 7.04 (4.44–11.60) |
| Bronchiolitis/RSV | 7.27 (3.83–14.50) | 2.01 (0.55–8.15) | 2.90 (0.64–20.40) | 5.82 (3.26–10.80) |
| Any Listed | 6.15 (4.21–9.05) | 3.07 (1.99–4.79) | 3.91 (2.55–6.12) | 4.50 (3.52–5.78) |

\*Adjusted for sex, underweight at birth, maternal smoking during pregnancy, breastfeeding, number of older siblings, SES, recruitment site and global ancestry. Recruitment site was not included in the African American models as they were recruited from one site only.

respectively, were 2.17 (95% CI: 1.12–4.24) and 4.77 (95% CI: 2.76–8.71) for URI, 2.05 (95% CI: 0.96–4.61) and 1.90 (95% CI: 0.97–3.86) for pneumonia, 4.78 (95% CI: 2.39–10.40) and 2.70 (95% CI: 1.08–7.71) for bronchitis, 2.01 (95% CI: 0.55–8.15) and 2.90 (95% CI: 0.64–20.40) for bronchiolitis/RSV, and 3.07 (95% CI: 1.99–4.79) and 3.91 (95% CI: 2.55–6.12) if the participant reported having at least one early-life respiratory illness (**Table 2 and Fig 3**).

To test statistically significant differences in odds ratios between racial/ethnic groups, we compared the beta coefficients using t-tests. [34] We found that there were significant differences ($p < 0.05$) between Puerto Ricans and Mexican Americans for URI and Any Listed, and between Puerto Ricans and African Americans for bronchitis and pneumonia (**S1 Table**).

In a separate analysis, we examined these associations including those individuals diagnosed with asthma before the age of two (**S2 Table**). Our results showed that all early-life respiratory illnesses were significantly associated ($p < 0.05$) with asthma diagnosis in our population and regardless of race/ethnicity, with the exception of RSV in Mexican Americans (**S3 Table**). It is worth noting that we observed significantly higher odds of asthma in Puerto Ricans than in Mexican Americans for every respiratory infection examined (**S4 Table**).

We additionally compared the odds of asthma after an early-life respiratory illness in Puerto Ricans living in Puerto Rico (Islanders) with those living in the mainland United States (Mainlanders); descriptive characteristics for these populations can be found in **S5 Table**. We found high asthma odds following early-life respiratory illness in Islanders relative to Mainlanders (**S6 Table**) though due to the small number of Mainlanders we were underpowered to detect statistically significant differences between these groups.

## Discussion

Overall, we found that early-life respiratory illnesses such as URI, pneumonia, bronchitis, and bronchiolitis/RSV are significantly associated with asthma diagnosis after the age of two regardless of race/ethnicity. However, Puerto Ricans had the strongest associations.

The association between early-life respiratory illnesses and asthma has been well documented. [14–24] Yet, little is known about racial/ethnic differences observed for the associations between early-life respiratory illnesses and the development of asthma later on in childhood.

Previous research on the association between early-life respiratory infection and the development of asthma has yielded conflicting results. Having a wheezing illness due to RSV or HRV infection in early life has previously shown to be associated with a 2.6- and 9.8-fold increase, respectively, in asthma odds by age six in a mostly white population. [18] Additionally, a recent study found that the severity of the RSV infection was strongly associated with childhood wheezing at age five in a mostly white population. [17] However, the vast majority

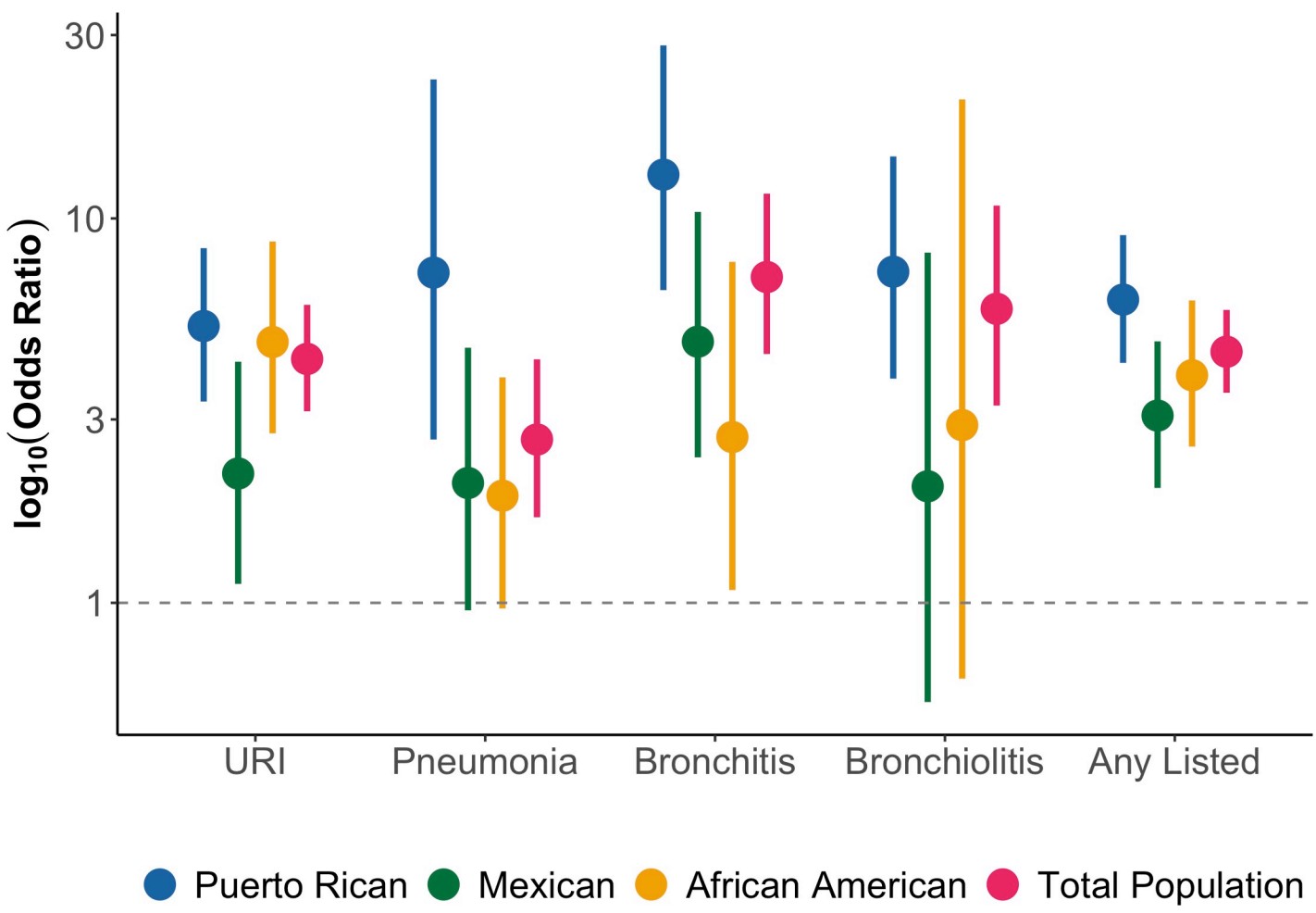

**Fig 3. Odds ratios plotted on a log scale of the association between early-life respiratory illnesses before the age of two and asthma diagnosis after the age of two according to race/ethnicity and in the total population.**

of children infected by respiratory viruses like RSV do not go on to develop respiratory illnesses like recurrent wheezing and asthma. [33] Currently, it is unclear why only a minority of children develop asthma after exposure to an early-life respiratory illness. One plausible explanation is that these respiratory illnesses alter the airway in early life, which leads to asthma later on in childhood. In fact, previous studies have shown that in adults with asthma, the airway epithelium is altered in a heterogeneous manner. [35] Another possibility is that children who are already genetically or environmentally prone to asthma present with early-life respiratory illness as an early manifestation of asthma. Indeed, previous studies have shown asymptomatic carriage of RSV and rhinovirus in children, [36–38] suggesting a spectrum of disease which may be affected by underlying asthma predisposition. [39, 40] Our findings show greater asthma odds associated with early-life respiratory illnesses in all populations studied, but that the odds was stronger in the Puerto Rican population. These findings are consistent with previous studies [40–42] showing that early-life respiratory illnesses are significantly associated with the development of asthma and asthma-related outcomes while additionally adding that these associations may vary by region and/or population.

Most children who develop respiratory viral infections in early life experience minor illness, but some develop much more severe illnesses that involve lower respiratory symptoms like

wheezing. [15, 16, 36] These more severe wheezing illnesses at an early stage in life are associated with a high risk for recurrent wheezing and asthma later in childhood, [18, 19] particularly if these wheezing illnesses were caused by RSV or HRV. [14, 17, 21–24] Puerto Rican children are especially at risk due to the year-round seasonality of RSV infections. [29, 30] In fact, 1,406 cases of bronchiolitis/RSV were reported in children with a mean age of seven months from six Puerto Rican hospitals over a period of nine months. [39] Our findings showed that not only is the prevalence of bronchiolitis/RSV infection highest in Puerto Rican children in our study population, but that the odds for asthma among children who have experienced a bronchiolitis/RSV infection in early life was the highest for Puerto Rican Islanders. These observations suggest that the burden of childhood asthma after bronchiolitis/RSV is particularly high Puerto Rican children. While it is known that early-life respiratory infections are associated with asthma, our findings show racial/ethnic disparities for the odds of asthma post-infection in our population, with Puerto Rican children carrying the greater burden of asthma.

In our study of children diagnosed with asthma after two years of age, some confidence intervals were too wide to make definitive conclusions regarding the odds ratios for asthma after infection in different racial/ethnic groups. However, when we included those children diagnosed with asthma before two years of age, we were able to better delineate the association between different respiratory infections and asthma development across racial/ethnic groups. Interestingly Puerto Ricans and Mexican Americans are both considered Hispanic/Latino for pulmonary function testing and other health outcomes. However, these populations had widely divergent prevalence of asthma after early-life respiratory infections. Puerto Ricans consistently had higher odds of asthma after all respiratory infections studied. Whether the observed association between early-life respiratory infection and greater asthma odds is causal of the alarmingly high prevalence of asthma in Puerto Ricans requires further study.

While we are not able to draw definitive conclusions about the greater odds of asthma after respiratory infection in Puerto Rican Islanders versus Mainlanders due to the limited number of mainland US Puerto Rican children studied, we find it interesting that Mainlander Puerto Ricans tended to have a lower odds of asthma after respiratory infections relative to Islanders. This may suggest an environmental effect associated with living in Puerto Rico that affects the prevalence of asthma. Respiratory infections likely represent one of several factors leading to increased asthma in Islander Puerto Ricans relative to Mainlanders, which may include other environmental and socio-cultural factors. Further studies examining this associations in Puerto Ricans living in the island and in the mainland US can help clarify this observation.

Our study may have been limited because the majority (71%) of Puerto Ricans in the GALA study were diagnosed with asthma before the age of two. In contrast, only 35% of Mexican Americans, and 50% of African Americans were diagnosed with asthma before two years old. Our secondary analysis including children diagnosed with asthma before two years of age addressed these differences. Additionally, our exposure measurements were determined retrospectively, which may introduce biases in the data. Specifically, recall bias may affect the accuracy of answers to questions about events in the distant past. However, if bias were to occur, this bias will be non-differential and affect all racial/ethnic groups studied, biasing the results towards the null. This study was underpowered to detect any significant associations of early-life respiratory illnesses on asthma in Puerto Rican Mainlanders versus Puerto Rican Islanders. Thus, we were not able to determine whether the year-round RSV prevalence in Puerto Rico influenced the association between respiratory illnesses and asthma. However, our study suggests that there are clear population-specific differences in asthma susceptibility across a variety of respiratory illnesses using a large cohort of minority children from three distinct racial/

ethnic populations. Moreover, our study also benefits from the wide range of clinical, social, and genetic data available on these children.

## Conclusion

Early-life respiratory illnesses such as RSV, which is highly prevalent in Puerto Ricans, have previously been associated with asthma. Our findings indicate that early-life respiratory infections are particularly associated with the later development of asthma in our population regardless of race/ethnicity, with Puerto Rican children having the stronger associations.

## Supporting information

**S1 Text. Derivation of socioeconomic status (SES).**
(DOCX)

**S2 Text. Derivation of global ancestry estimates.**
(DOCX)

**S1 Table. Pairwise comparison of odds ratios for asthma after two years of age following early-life respiratory infection by racial/ethnic group.** *Definition of Abbreviations*:
URI = Upper Respiratory Infection.
(DOCX)

**S2 Table. Descriptive statistics for selected characteristics of the study population including those diagnosed with asthma before age two (N = 3,722).**
(DOCX)

**S3 Table. Odds ratios and confidence intervals from the population-specific and combined association analysis between respiratory illnesses in the first two years of life and physician-diagnosed asthma according to race/ethnicity and in the total population in GALA II and SAGE: 2006–2014.** *Definition of Abbreviations*: URI = Upper Respiratory Infection.
(DOCX)

**S4 Table. Pairwise comparison of odds ratios for asthma following early-life respiratory infection by racial/ethnic group including those diagnosed with asthma before age two.**
(DOCX)

**S5 Table. Descriptive characteristics of Puerto Rican Islanders and Mainlanders.**
(DOCX)

**S6 Table. Odds ratios and confidence intervals from the geography-specific analysis between respiratory illnesses in the first two years of life and physician-diagnosed asthma after the age of two in the two Puerto Rican populations: Islanders and Mainlanders.** *Definition of Abbreviations*: URI = Upper Respiratory Infection.
(DOCX)

## Acknowledgments

The authors acknowledge the families and patients for their participation and thank the numerous health care providers and community clinics for their support and participation in GALA II / SAGE. In particular, the authors thank study coordinator Sandra Salazar; the recruiters who obtained the data: Duanny Alva, MD, Gaby Ayala-Rodríguez, Lisa Caine, Elizabeth Castellanos, Jaime Colón, Denise DeJesus, Blanca López, Brenda López, MD, Louis

Martos, Vivian Medina, Juana Olivo, Mario Peralta, Esther Pomares, MD, Jihan Quraishi, Johanna Rodríguez, Shahdad Saeedi, Dean Soto, Ana Taveras.

## Author Contributions

**Conceptualization:** Luisa N. Borrell, Max A. Seibold, José R. Rodríguez-Santana, Esteban G. Burchard.

**Data curation:** Jennifer R. Elhawary, Celeste Eng, Sandra Salazar.

**Formal analysis:** Luisa N. Borrell, Jennifer R. Elhawary, Sam S. Oh, Esteban G. Burchard.

**Investigation:** Luisa N. Borrell, Celeste Eng, Sandra Salazar, Max A. Seibold, José R. Rodríguez-Santana, Esteban G. Burchard.

**Methodology:** Eric M. Wohlford, Luisa N. Borrell, Jennifer R. Elhawary, Sam S. Oh, Esteban G. Burchard.

**Project administration:** Esteban G. Burchard.

**Supervision:** Luisa N. Borrell, Sam S. Oh, Esteban G. Burchard.

**Visualization:** Jennifer R. Elhawary.

**Writing – original draft:** Eric M. Wohlford, Luisa N. Borrell, Jennifer R. Elhawary, Sam S. Oh, Esteban G. Burchard.

**Writing – review & editing:** Eric M. Wohlford, Luisa N. Borrell, Jennifer R. Elhawary, Brian Plotkin, Sam S. Oh, Thomas J. Nuckton, Michael A. LeNoir, Kelley Meade, Harold J. Farber, Denise Serebrisky, Emerita Brigino-Buenaventura, William Rodriguez-Cintron, Rajesh Kumar, Shannon Thyne, Max A. Seibold, José R. Rodríguez-Santana, Esteban G. Burchard.

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
