## [Decision Letter · Decision Letter 0]

4 Feb 2020

PONE-D-19-32435

Differential asthma risk following respiratory infection in children from three minority populations

PLOS ONE

Dear Dr. Borrell,

Thank you for submitting your manuscript to PLOS ONE. After careful consideration, we feel that it has merit but does not fully meet PLOS ONE’s publication criteria as it currently stands. Therefore, we invite you to submit a revised version of the manuscript that addresses the points raised during the review process.

We would appreciate receiving your revised manuscript by Mar 20 2020 11:59PM. To enhance the reproducibility of your results, we recommend that if applicable you deposit your laboratory protocols in protocols.io, where a protocol can be assigned its own identifier (DOI) such that it can be cited independently in the future. For instructions see: http://journals.plos.org/plosone/s/submission-guidelines#loc-laboratory-protocols

We look forward to receiving your revised manuscript.

Kind regards,

Sreekumar Othumpangat, PhD

Academic Editor

PLOS ONE

Additional Editor Comments (if provided):

In this manuscript authors address whether early life respiratory illness was associated with later life asthma in minority children. There are several concerns that need to be addressed as stated by reviewer 1 and 2.

Authors need to add more information on bronchiolitis of RSV and HRV. It is not clear whether the self-reported asthma or diagnosed by a physician before the age of 2 from the methods presented.

In results the Puerto Ricans were more likely to report RSV infection than the other minorities, does this has anything to do with the genetics or is it fully related to the earlier infection?

Not much information provided with the early life HRV, does the data show a similar effect as RSV?

Discussion may address other factors like environment, genetic variation or life style that could have a role in mainlanders and the islanders.

Line 298 to 302, modify the sentences, to avoid confusion on mainlanders and islanders.

Journal Requirements:

2. Please provide additional details regarding participant consent. In the ethics statement in the Methods and online submission information, please ensure that you have specified whether consent was written or verbal/oral. If consent was verbal/oral, please specify: 1) whether the ethics committee approved the verbal/oral consent procedure, 2) why written consent could not be obtained, and 3) how verbal/oral consent was recorded. If your study included minors, please state whether you obtained consent from parents or guardians in these cases.

3. Thank you for including your ethics statement:  

"All participating institutions obtained the appropriate approvals from their Institutional Review Boards."

Reviewers' comments:

Reviewer's Responses to Questions

**Comments to the Author**

1. Is the manuscript technically sound, and do the data support the conclusions?

Reviewer #1: Partly

Reviewer #2: Yes

2. Has the statistical analysis been performed appropriately and rigorously? 

Reviewer #1: I Don't Know

Reviewer #2: Yes

3. Have the authors made all data underlying the findings in their manuscript fully available?

Reviewer #1: Yes

Reviewer #2: No

4. Is the manuscript presented in an intelligible fashion and written in standard English?

Reviewer #1: Yes

Reviewer #2: Yes

5. Review Comments to the Author

Reviewer #1: The authors present an analysis looking at the relationship between ethnicity and asthma, using 2006 through 2014 data from two case-control studies (GALA II and SAGE). The authors report that the odds of asthma were higher among Puerto Ricans (from the island of Puerto Rico, not the mainland USA) than the odds observed among Mexican Americans and African Americans.

This study is overall interesting and has scientific merit. There are several issues that the authors need to address and clarifications that are needed, in order to better understand the findings. One big limitation of the study is that it relies upon self-reporting of history of respiratory infections as well as diagnosis of asthma, both of which are subject to recall bias among other biases.

Please refer to attached document for additional recommendations.

Reviewer #2: In this work, the authors explored the association between early-life respiratory illness and physician-diagnosed asthma at 2 years in three minority populations and found a strong association. Though the findings are not new, it has strengthened existing evidence. The authors need to clearly mention the value of the findings and specific recommendations for the next step.

Specific comments and suggestions:

1. It would be worth to mention in the background some hypothesis citing previous relevant work why asthma morbidity and mortality has been strikingly higher among Puerto Rican as early-life respiratory infections are one of many drivers that contributes asthma development

2. It is not clear whether the etiologies of respiratory illnesses (URI, pneumonia, bronchitis) were investigated. Most published works so far reportedly linked early life viral respiratory infections with childhood asthma. If the etiologic data is available it would be good to perform a sub-analysis to look for specific association between viral URI/pneumonia before two years and asthma at two years of age and how this varies by ethnicity

3. Line 151: It is not clear whether all bronchiolitis episodes had a detectable RSV, although we know that most bronchiolitis is caused by RSV

4. Line 154: atopic status (this might vary by ethnicity) of enrolled children needed to be addressed as a covariate as this might influence the relationship between respiratory infections and risk of asthma development (https://www.sciencedirect.com/science/article/abs/pii/S0091674907002382

6. PLOS authors have the option to publish the peer review history of their article (what does this mean?). If published, this will include your full peer review and any attached files.

Reviewer #1: No

Reviewer #2: Yes: Md. Zakiul Hassan

---

## [Editor Report · Decision Letter 1]

1 Apr 2020

Differential asthma odds following respiratory infection in children from three minority populations

PONE-D-19-32435R1

Dear Dr. Borrell,

We are pleased to inform you that your manuscript has been judged scientifically suitable for publication and will be formally accepted for publication once it complies with all outstanding technical requirements.

With kind regards,

Sreekumar Othumpangat, PhD

Academic Editor

PLOS ONE
---

## [Editor Report · Acceptance letter]

6 Apr 2020

PONE-D-19-32435R1 

Differential asthma odds following respiratory infection in children from three minority populations 

Dear Dr. Borrell:

I am pleased to inform you that your manuscript has been deemed suitable for publication in PLOS ONE. Congratulations! Your manuscript is now with our production department. 

With kind regards,

on behalf of

Dr. Sreekumar Othumpangat 

Academic Editor

PLOS ONE